# Chromophore carbonyl twisting in fluorescent biosensors encodes direct readout of protein conformations with multicolor switching

Malin J. Allert [1], Shivesh Kumar[1,2], You Wang[1], Lorena S. Beese[1] & Homme W. Hellinga [1✉]

Fluorescent labeling of proteins is a powerful tool for probing structure-function relationships with many biosensing applications. Structure-based rules for systematically designing fluorescent biosensors require understanding ligand-mediated fluorescent response mechanisms which can be challenging to establish. We installed thiol-reactive derivatives of the naphthalene-based fluorophore Prodan into bacterial periplasmic glucose-binding proteins. Glucose binding elicited paired color exchanges in the excited and ground states of these conjugates. X-ray structures and mutagenesis studies established that glucose-mediated color switching arises from steric interactions that couple protein conformational changes to twisting of the Prodan carbonyl relative to its naphthalene plane. Mutations of residues contacting the carbonyl can optimize color switching by altering fluorophore conformational equilibria in the apo and glucose-bound proteins. A commonly accepted view is that Prodan derivatives report on protein conformations via solvatochromic effects due to changes in the dielectric of their local environment. Here we show that instead Prodan carbonyl twisting controls color switching. These insights enable structure-based biosensor design by coupling ligand-mediated protein conformational changes to internal chromophore twists through specific steric interactions between fluorophore and protein.

[1] Department of Biochemistry, Duke University Medical Center, Durham, NC 27710, USA. [2] Present address: Department of Biochemistry and Molecular Biophysics, Washington University in St. Louis, St. Louis, MO 63110, USA. ✉email: hwh@biochem.duke.edu

Determination of distributions of protein conformations as a function of ligand concentration, solvent composition, physical conditions, and time is central to understanding protein structure-function relationships[1,2]. Labeling proteins with fluorescent probes that respond to protein conformations is a powerful tool for measuring such conformational distributions in vitro and in vivo, and has many applications in cell biology, neurobiology, clinical chemistry, and environmental chemistry[3]. Fluorescent probes are deployed most effectively if the mechanisms by which their responses are coupled to protein conformations are understood in sufficient detail to enable their installation into proteins by application of structural principles[4].

Color switching arises when a fluorescent system exchanges emission intensity at two distinct wavelengths in response to protein conformational changes. Such exchanges enable robust readout of conformational distributions, because they enable quantitation by ratiometry, thereby eliminating signal fluctuations due to protein concentration variability, and intensity attenuation[5,6]. Prodan derivatives are archetypes for measuring protein conformational changes via color switching (Fig. 1). They have been used to probe protein conformational distributions in vitro using thiol-reactive derivatives (Fig. 1c, d) and in vivo with non-natural amino acids (Fig. 1e, f) to study the hydrophobic nature of protein cavities[7,8], detect ligand binding in biosensors[9-15], screen for enzyme inhibitor leads[16-22], detect protein-protein interactions[23,24], characterize membrane protein conformational changes in vivo[24-27], measure transmembrane voltages[28], and analyze complex thermal folding and ligand-binding landscapes[29,30]. Despite their extensive use, effective placement of responsive Acrylodan derivatives relies largely on trial-and-error, rather than precise, structure-based design principles[4].

We have constructed conjugates of the *Escherichia coli* periplasmic glucose-binding protein (ecGBP) in which the thiol-reactive Prodan derivatives Acrylodan[31] and Badan (Fig. 1c, d) were installed singly at cysteine mutants within the glucose-binding site, replacing a tryptophan or phenylalanine side-chain that directly contacts the bound glucose (Fig. 1a). The ecGBP.183C•Acrylodan conjugate[9,29,30] has been used for continuous glucose monitoring in animals[32] and humans[33]. We have also identified thermostable color-switching homologs from *Thermoanaearobacterium thermosaccharyliticum* (ttGBP) and *Geobacillus kaustophilus* (gkGBP)[34].

We report spectroscopic, structural, and mutagenesis studies which revealed that glucose-mediated color switching in these GBP conjugates exhibit exchanges between four excited-state emissions. These four colors switch as two independent pairs, comprising a dominant blue↔green and minor violet↔cyan switch. Individual conjugates differ in their color-switching direction of the dominant pair: blue apo-protein→green glucose complex, or vice versa. The absorbance spectra also exhibited glucose-dependent exchanges between paired absorption bands, indicating that fluorophore ground states play an important role in color switching. Two X-ray structures representing conjugates in the ground states of green- and blue-emitting forms revealed that color switching is associated with twisting of the fluorophore carbonyl in or out of the naphthalene ring plane, respectively. This rotatable carbonyl (Fig. 1b) is buried entirely by a small pocket within each protein where it is grasped by two or three residues. Each naphthalene ring is held within a short channel

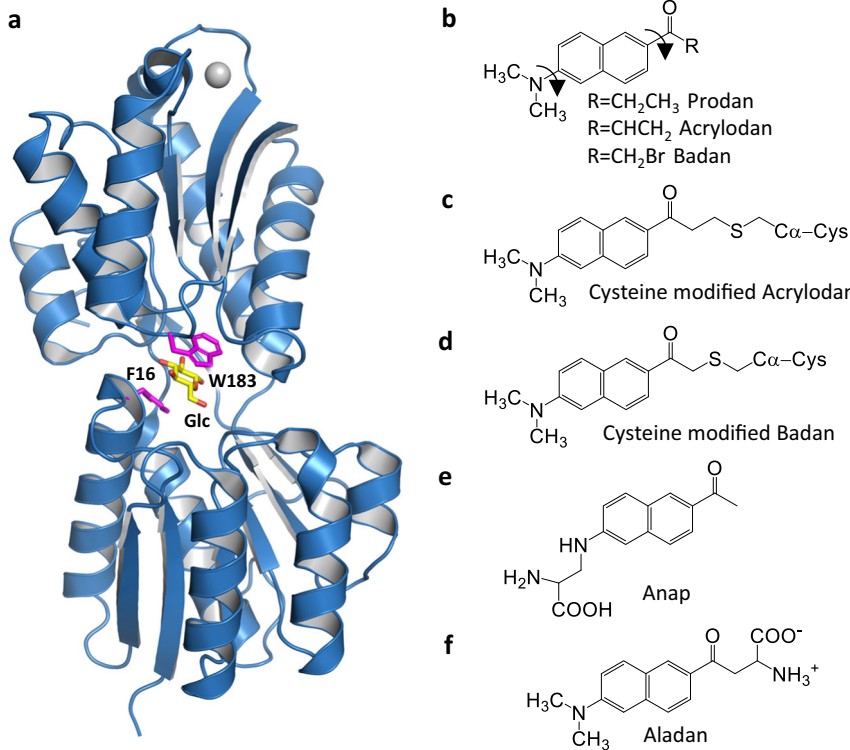

**Fig. 1 *E. coli* glucose-binding protein and Prodan derivatives. a** ecGBP. The two aromatic residues (magenta) form van der Waals contacts with opposing pyranose ring faces of bound glucose (Glc) (yellow). Gray circle, $Ca^{2+}$. **b** Prodan and its derivatives. Arrows indicate rotatable bonds of the functional groups bearing lone pairs. *R* groups for thiol-reactive derivatives are shown (the conjugated Acrylodan linker is one methylene unit longer than that of Badan). For membrane probes, *R* is a lipid. **c** Conjugated Acrylodan. The thiol sulfur reacts with the terminal $CH_2$ in the acryloyl reactive group. **d** Conjugated Badan. The thiol replaces the bromine. **e** In the non-natural amino acid Anap the Prodan core is incorporated as a side-chain that links to the amino acid backbone to the naphthalene via a secondary amine replacing the original dimethyl amino group. **f** The non-natural amino acid Aladan couples the Prodan core via the carbonyl.

and partially protrudes into the solvent. Mutagenesis studies established that the pocket residues determine the direction and degree of color change. The steric interactions between the protein and fluorophore vary with glucose-mediated protein conformational changes and elicit a color switch by torqueing the carbonyl relative to the naphthalene ring. Apparent spectral color changes then arise from the equilibrium distribution of the two dominant blue and green spectral states at different glucose concentrations. Based on these insights, we propose a conceptually straightforward structure-based framework for designing biosensors by coupling ligand-mediated protein conformational changes to internal fluorophore twists through specific steric interactions between fluorophore and protein.

## Results and discussion

**Construction of fluorescent glucose-binding protein conjugates**. Glucose is bound at the interface between two domains of ecGBP (Fig. 1a)[35–42]. The protein adopts two conformations that interconvert via bending of a three-stranded β hinge connecting these domains. In the absence of glucose, the protein adopts an open conformation. Glucose binding stabilizes a closed conformation in which the sugar pyranose ring is sandwiched between two aromatic side-chains and surrounded by an annulus of hydrogen bonds between side-chains and glucose hydroxyls.

Acrylodan or Badan were attached to unique cysteines that replaced either the tryptophan or the phenylalanine residues flanking the bound glucose in ecGBP and the ttGBP (Fig. 1a). The ecGBP.W183C, ttGBP.W182C are homologous, as are ecGBP.F16C and ttGBP.F17C[34].

**Glucose-mediated color switching is bidirectional and biphasic**. We used an LED light source with broad excitation (365 nm ± 10 nm) to measure fluorescence emission spectra as a function of glucose concentration in Acrylodan and Badan conjugates that replaced one of the two aromatic sandwich residues in three GBP homologs (Fig. 2; Table 1). Analysis of intensity-weighted dominant emission wavelengths, barychrome $\lambda_B$ (Eq. 1), revealed four color-switching patterns upon glucose binding: Green→Blue (Fig. 2a, b); Blue→Green (Fig. 2c); Green→Yellow (Fig. 2d, ecGBP.183C•Badan); no color change (Fig. 2c, ttGBP.182C•Badan, columns 2 and 3). Color changes therefore can be either hypsochromic (blue shift) or bathochromic (red shift), depending on the conjugate. The relative intensities of the entire emission spectra, $\rho$ (Eq. 5), also are correlated with apparent spectral color: Green is less intense than Blue, whereas Green and Yellow are approximately equally intense. A two-state, single-site glucose-binding isotherm (Eq. 2) accurately modeled the glucose dependence of both $\lambda_B$ and $\rho$ (Fig. 2, columns 2

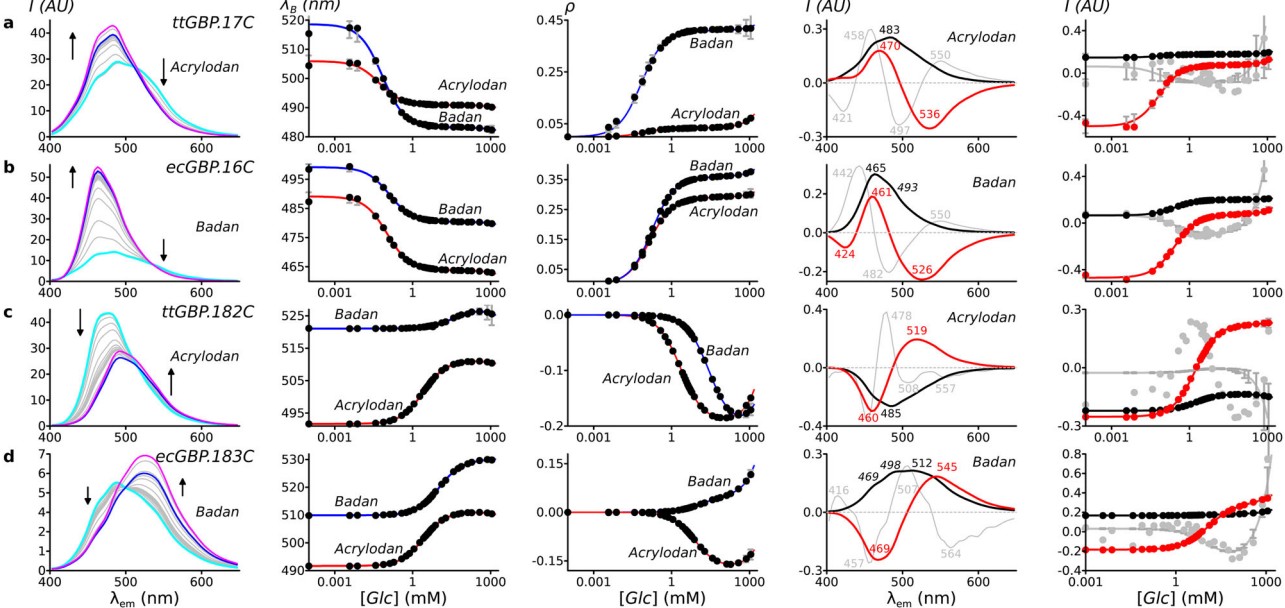

**Fig. 2 Changes in fluorescence emission spectra in response to glucose binding.** The selected conjugates represent three main apparent color switching patterns upon binding glucose (Glc). Apparent colors were classified according to the spectral barychrome[34], $\lambda_B$, value: Blue < 500 nm ; 500 nm < Green < 520 nm: Yellow >520 nm. **a** Green→Blue, ttGBP.17C; spectral analysis for ttGBP.17C•Acrylodan is shown in detail (note that the ttGBP.17C•Badan exhibits a larger change). **b** Green→Blue, ecGBP.16C; analysis for Badan conjugate is detailed. **c** Blue→Green, ttGBP.182C; Acrylodan conjugate is detailed (ttGBP.182C•Badan conjugate is non-responsive). **d** Green→Yellow, ecGBP.183C; Badan conjugate is detailed (response of ecGBP.183C•Acrylodan conjugate is similar to ttGBP.182C•Acrylodan). Column 1, emission spectra (conjugated fluorophore is indicated; arrows: direction of intensity changes upon glucose addition; cyan: apo-protein spectrum; blue, spectrum at [Glc]≈5x$K_d$; magenta: spectrum at maximal [Glc]). Column 2, apparent color switching monitored by $\lambda_B$. Solid lines: binding isotherm fit using linear baselines for the glucose complex; circles: experimental data. Column 3, change in the logarithm of total emission intensity relative to the apo-protein spectrum, $\rho$ ($\rho$=0 in the absence of glucose; positive, intensity increases with glucose addition; negative intensity decreases); solid lines: binding isotherm fit with linear baselines for the glucose complex; circles: experimental data; the variations in the fit models (± one standard deviation), calculated from 10,000 bootstrap trials, are shown as vertical gray lines at each experimental data point (distributions may be smaller than the data point circles). Column 4, top three spectral components of the SVD of emission spectra glucose dependence (color-coded numbers indicate peak or shoulder wavelengths; positive peak, increases with glucose; negative peak, decreases with glucose): black, $C_1$, largely invariant; red, $C_2$, the most glucose-responsive spectral component (peaks of opposite signs mark electronic transitions that exchange in response to glucose binding); gray, $C_3$, the component primarily encoding the solvatochromic response at high glucose concentrations. Column 5, binding isotherm fits of the glucose-dependent contributions of the top three components, using a linear baseline for the glucose complex; the three isotherms have identical $K_d$ values. For each component, 1σ variations are shown as gray vertical lines. Additional spectra and SVD decompositions are shown in Supplementary Figs 1–4.

## Table 1 Properties of LED-excited emission spectra determined from glucose titrations[a].

| Conjugate[b] | | | Barymetry[c] | | | | $^{B}K_d$ (mM) | Intensity changes[d] $\rho^*$ | SVD components[e] | | | | | | | |
| Protein | Fluorophore | Mutation | Apo (nm) | Bound (nm) | $\Delta\lambda_{AB}$ (nm) | Color switch | | | C1 b | C1 c | C1 g | C2 v | C2 b | C2 c | C2 g | C2 y |
|---|---|---|---|---|---|---|---|---|---|---|---|---|---|---|---|---|
| ecGBP.W183C | A | H152N | 493 ± 0.0 | 511 ± 0.0 | 17 ± 0.0 | B → G | 8.2 ± 0.0 | −0.17 ± 0.00 | | + | | | + | | + | |
| | | H152A | 499 ± 0.2 | 511 ± 3.0 | 12 ± 3.2 | B → G | 20 ± 7.5 | −0.15 ± 0.02 | | + | | | + | | + | |
| | | H152A | 515 ± 0.0 | 521 ± 0.2 | 6 ± 0.3 | Ø | 2.7 ± 0.5 | 0.08 ± 0.00 | | | | | | | + | |
| | | H152Q | 505 ± 0.0 | 516 ± 0.3 | 11 ± 0.3 | G → G | 24 ± 0.7 | −0.07 ± 0.00 | | + | + | | + | | + | |
| | | H152F | 499 ± 0.0 | 516 ± 3.5 | 17 ± 3.5 | B → G | 31 ± 7.5 | −0.10 ± 0.02 | | + | | | + | | + | |
| | | K92A | 491 ± 0.0 | 473 ± 0.6 | −18 ± 0.6 | B → B | 100 ± 3.5 | 0.09 ± 0.01 | | + | | | + | + | + | + |
| | | K92M | 489 ± 0.0 | 488 ± 0.1 | −1 ± 0.1 | Ø | 72 ± 6.7 | −0.07 ± 0.00 | | + | | | + | + | + | |
| | | E149Q | 497 ± 0.0 | 510 ± 0.1 | 13 ± 0.1 | B → G | 0.51 ± 0.01 | −0.09 ± 0.00 | | + | + | | + | + | + | + |
| | | E149S | 500 ± 0.3 | 524 ± 1.9 | 25 ± 2.2 | G → Y | 0.53 ± 0.12 | −0.15 ± 0.00 | | + | | | + | + | + | + |
| | | E149K | 497 ± 0.0 | 517 ± 1.5 | 20 ± 1.5 | B → G | 1.8 ± 0.2 | −0.17 ± 0.02 | | + | | | + | | + | |
| | B | H152N | 510 ± 0.0 | 530 ± 0.0 | 20 ± 0.0 | G → Y | 11 ± 0.0 | 0.05 ± 0.00 | | | + | + | | | | |
| | | H152A | 510 ± 0.1 | 521 ± 4.1 | 11 ± 4.2 | G → G | 120 ± 62 | 0.19 ± 0.09 | | + | + | + | + | + | + | |
| | | H152Q | 521 ± 0.1 | 532 ± 4.3 | 11 ± 4.4 | G → G | 19 ± 14 | 0.09 ± 0.03 | | | | | + | | | |
| | | K92A | 519 ± 0.0 | 525 ± 2.9 | 6 ± 3.0 | Y → Y | 93 ± 53 | 0.01 ± 0.01 | | + | + | | + | | + | |
| | | K92M | 512 ± 0.0 | 508 ± 1.2 | −4 ± 1.2 | G → Y | 130 ± 71 | 0.07 ± 0.00 | | | | | | | | |
| ttGBP.W182C | A | H151A | 492 ± 0.0 | 511 ± 0.0 | 20 ± 0.0 | B → G | 2.5 ± 0.0 | −0.19 ± 0.00 | | + | | | + | + | + | |
| | | H151N | 498 ± 0.0 | 516 ± 1.1 | 19 ± 1.1 | B → G | 18 ± 1.3 | −0.17 ± 0.02 | | + | | | + | + | + | |
| | | H151Q | 495 ± 0.0 | 515 ± 0.2 | 20 ± 0.3 | B → G | 1.9 ± 0.0 | −0.21 ± 0.00 | | + | | | + | + | + | |
| | | H151Q | 503 ± 0.0 | 517 ± 0.1 | 14 ± 0.1 | G → G | 0.48 ± 0.02 | −0.08 ± 0.00 | | + | | | + | + | + | |
| | | R91A | 489 ± 0.0 | 517 ± 1.7 | 29 ± 1.7 | B → G | 6.8 ± 0.6 | −0.29 ± 0.04 | | + | | | + | + | + | |
| | | R91M | 489 ± 0.0 | 515 ± 1.4 | 26 ± 1.5 | B → G | 5.8 ± 0.5 | −0.29 ± 0.00 | | + | | | + | + | + | |
| | | R91K | 495 ± 0.7 | 506 ± 0.2 | 12 ± 0.9 | B → G | 1.1 ± 0.2 | −0.08 ± 0.00 | | + | | | + | + | + | |
| | | R91S | 489 ± 0.1 | 518 ± 0.8 | 30 ± 0.9 | B → G | 6.8 ± 0.2 | −0.37 ± 0.03 | + | + | | | + | + | + | |
| | | Q148E | 495 ± 0.1 | 521 ± 0.1 | 27 ± 0.2 | B → Y | 0.78 ± 0.01 | −0.24 ± 0.00 | + | + | + | | + | + | + | |
| | | R91K, Q148E | 488 ± 0.0 | 519 ± 0.1 | 32 ± 0.1 | Y → Y | 9.4 ± 0.0 | −0.35 ± 0.00 | | + | | | + | + | + | |
| | B | | 521 ± 0.0 | 527 ± 0.0 | 6 ± 0.2 | Y → Y | 36 ± 2.1 | −0.20 ± 0.00 | | + | | | + | | + | |
| ecGBP.F16C | A | N256A | 489 ± 0.4 | 464 ± 0.0 | −25 ± 0.5 | B → B | 0.1 ± 0.00 | 0.29 ± 0.00 | + | + | | + | + | | + | |
| | | N256L | 494 ± 0.0 | 482 ± 0.0 | −11 ± 0.0 | B → B | 10 ± 0.1 | 0.11 ± 0.00 | + | + | | + | + | | + | |
| | | N256D | 492 ± 0.0 | 484 ± 0.2 | −7 ± 0.2 | B → B | 17 ± 0.8 | 0.15 ± 0.00 | | + | + | | + | | + | |
| | | N256D | 489 ± 0.0 | 486 ± 0.0 | −3 ± 0.0 | Ø | 18 ± 0.5 | 0.11 ± 0.00 | + | + | | | + | | + | |
| | | D236A | 490 ± 0.0 | | | | nb | | | | | | | | | |
| | | D236L | 492 ± 0.0 | | | | nb | | | | | | | | | |
| | | D236N | 489 ± 0.0 | 486 ± 0.0 | −3 ± 0.0 | G → B | 18 ± 0.5 | 0.11 ± 0.00 | | | | | | | | |
| | B | N256A | 499 ± 0.4 | 480 ± 0.1 | −19 ± 0.4 | G → B | 0.13 ± 0.00 | 0.36 ± 0.00 | + | + | | + | + | + | + | |
| | | N256L | 499 ± 0.0 | 487 ± 0.1 | −12 ± 0.2 | G → B | 5.6 ± 0.3 | 0.25 ± 0.00 | + | + | | + | + | + | + | |
| | | N256D | 498 ± 0.0 | 496 ± 0.2 | −2 ± 0.2 | Ø | 15 ± 2.1 | 0.06 ± 0.00 | | + | | + | + | + | + | |
| | | D236A | 496 ± 0.2 | 497 ± 0.2 | 0 ± 0.4 | Ø | 28 ± 26 | 0.05 ± 0.01 | | + | | | + | | + | |
| | | D236L | 503 ± 4.8 | | | | nb | | | | | | | | | |
| | | D236N | 504 ± 0.0 | | | | nb | | | | | | | | | |
| ttGBP.F17C | A | N258A | 496 ± 0.0 | 497 ± 0.2 | 0 ± 0.2 | Ø | 27 ± 23 | 0.05 ± 0.01 | | + | | + | + | | + | |
| | | N258L | 506 ± 0.6 | 491 ± 0.0 | −15 ± 0.7 | G → B | 0.052 ± 0.004 | 0.03 ± 0.00 | | + | | | + | | + | |
| | | N258D | 517 ± 0.1 | 497 ± 0.1 | −19 ± 0.2 | G → B | 0.82 ± 0.04 | 0.16 ± 0.00 | + | + | + | | + | | + | |
| | | N258D | 514 ± 0.2 | 495 ± 0.1 | −19 ± 0.2 | G → B | 0.73 ± 0.02 | 0.14 ± 0.00 | | + | + | | + | | + | |
| | | D238A | 505 ± 0.0 | 503 ± 0.1 | −2 ± 0.1 | Ø | 28 ± 1.0 | 0.00 ± 0.00 | | + | | | + | | + | |
| | | D238L | 508 ± 0.0 | | | | nb | | | | | | | | | |
| | | D238N | 504 ± 0.0 | 502 ± 0.3 | −1 ± 0.3 | Ø | 85 ± 31 | −0.01 ± 0.00 | | | | | | | | |
| | B | D238N | 518 ± 0.0 | 490 ± 0.0 | −29 ± 0.0 | G → B | 7.5 ± 0.0 | 0.21 ± 0.00 | | + | | | + | + | + | |
| | | N258A | 519 ± 1.1 | 483 ± 0.1 | −35 ± 1.2 | Y → B | 0.6 ± 0.02 | 0.41 ± 0.00 | | + | + | + | + | | + | |
| | | N258L | 525 ± 0.3 | 489 ± 0.1 | −36 ± 0.3 | Y → B | 0.6 ± 0.004 | 0.53 ± 0.01 | | + | | | + | | + | + |
| | | N258D | 528 ± 0.0 | 491 ± 0.0 | −37 ± 0.1 | Y → B | 1.7 ± 0.01 | 0.42 ± 0.00 | | + | | | + | | + | + |
| | | D238A | 520 ± 0.0 | 527 ± 0.2 | 7 ± 0.3 | Y → G | 25 ± 1.3 | 0.02 ± 0.01 | | + | | | + | + | + | |
| | | D238L | 516 ± 0.0 | | | | nb | | | | | | | | | |
| | | D238N | 518 ± 0.2 | | | | nb | | | | | | | | | |
| | | | 523 ± 0.0 | 492 ± 0.0 | −30 ± 0.1 | Y → B | 0.89 ± 0.006 | 0.37 ± 0.00 | | + | | + | + | + | + | + |

[a]Individual spectra and their analyses are presented in Supplementary figures 1–4. [b]A, Acrylodan; B, Badan. [c]Barychromes of individual spectra were calculated according to Eq. 1. Titration series were fit with Eq. 2 (constant baseline for glucose complex) from which Apo and Bound barychromes, and glucose affinities, $^{B}K_d$ values (nb, no binding) were determined. Classification scheme of apparent color changes: B, blue (<500 nm); G, green (500 nm ≤ G <520 nm); Y, yellow (≥520 nm); Ø, no color change. [d]The relative changes in spectral intensity, $\rho$, was determined using Eq. 6. In the absence of glucose, $\rho^* = 1$, $\rho >1$: intensity decreases with glucose addition; $\rho^* >1$, intensity increases. [e]SVD analysis was calculated according to Eq. 7. The wavelengths of peaks in each component were determined as minima or maxima in first derivatives. Each peak was assigned to one of five electronic transitions (v, b, c, g, y) according to its wavelength: violet, v- 400–440 nm; blue, b: 440–470; cyan, c:470–500 nm; green, g: 500–540 nm; yellow, y: 540–590 nm. The presence or absence of these transitions is tabulated for each conjugate.

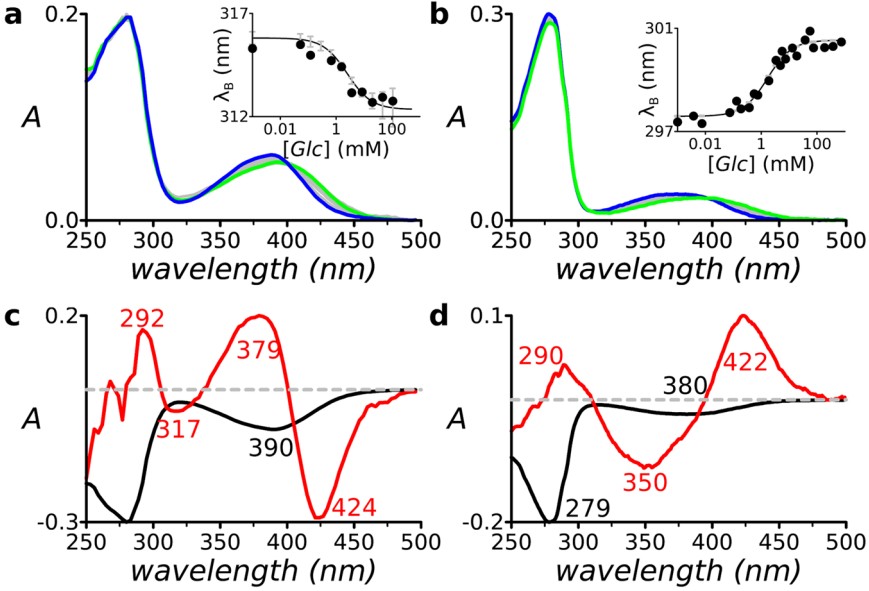

**Fig. 3 Glucose-dependent changes in absorption spectra of two conjugates that undergo hypso- or bathochromic shifts, respectively.**
**a** ecGBP.16C•Badan (apo-protein: green; glucose complex: blue). **b** ttGBP.182C•Acrylodan (apo-protein: blue; glucose complex: green), respectively. Inserts show dependence of apparent color on glucose, as reported by the barychrome. Lines: fit to single-site glucose-binding isotherm; circles: experimental data; vertical gray lines: 1σ variance of 10,000 bootstrap models at each data experimental glucose concentration. **c** and **d** Corresponding SVD analysis. The $C_1$(black) and $C_2$ (red) spectral components contained the color-switching information.

and 3). The conjugated chromophores therefore adopt two emission states in the apo (open) and glucose-bound (closed) protein conformations, respectively, with intermediate apparent spectral colors arising from mixtures as the protein population shifts in response to glucose binding.

At high glucose concentrations (>150 mM) the closed state population dominates, and each conjugate, regardless of color, undergoes small hypsochromic shifts and intensity increases that both are linearly dependent on glucose. Color switching therefore is biphasic: a dominant, bidirectional color and intensity switch associated with ligand binding, and minor hypsochromic shifts and intensity increases at high glucose concentrations.

**Color switching in response to glucose involves exchanges between paired emissions.** The electronic transitions that give rise to each apparent color are spaced closely together and difficult to distinguish within the broad peaks of the emission spectra. We therefore used singular value decomposition (SVD)[43] to identify the dominant wavelengths at which intensities change in response to glucose binding in 12- or 24-point titration series (Eq. 7). Our SVD analysis decomposes the change in wavelength-dependent intensities in response to a concentration series of glucose or other titrants into (i) a set of spectral components, $C_i$ (ii) the contributions of each component as a function of titrant concentration (Eq. 8) which were fit to isothermal ligand-binding models, and (iii) a single global weight for each component. The overall information content contribution of these spectral components to the emission spectra are ranked by their global weights ($C_1$ encodes a larger share of the overall intensity than $C_2$, etc.).

For all conjugates, the $C_1$–$C_3$ SVD spectral components encoded the color-switching information (Fig. 2, column 4; Supplementary Figs. 1–4); $C_{>3}$ encoded experimental noise. The $C_1$ contributions exhibited the smallest variation with glucose. $C_1$ therefore encodes emissions that are present in both apo and glucose-bound complexes. The $C_2$ contributions changed most over the concentration interval within which glucose binding occurs and contained paired regions that changed sign.

$C_2$ therefore encodes the wavelength-dependent emission exchange(s). The $C_3$ contributions were dominant only at glucose concentrations above which binding is saturated. $C_3$ therefore encodes the high-glucose phase of the responses. The two-state binding isotherm also accurately modeled the glucose dependence of the $C_2$ contributions (Fig. 2, column 5), consistent with this component encoding the two-state color switch.

Emissions grouped into five categories: violet, 400–440 nm; blue, 440–470 nm; cyan, 470–500 nm; green, 500–540 nm; yellow, 540–590 nm (Fig. 2). Blue, cyan, and green or yellow emissions were ubiquitous. The violet emission was identified only in ecGBP.16C•Badan (Fig. 2b, 424 nm region). The yellow emission was identified in ecGBP.W183C•Badan (Fig. 2d, 545 nm region) and several mutants (see below). In all conjugates, color switching was dominated by a dual emission that exchanges blue↔green/yellow regions (regions with opposing signs in $C_2$) in either direction (*cf*. Fig. 2a or b with Fig. 2c). In ecGBP.F16C•Badan, a second, minor, independent dual emission exchanged violet→cyan regions (Fig. 2b).

**Absorbance spectra also undergo color switching.** Below 300 nm tryptophan absorbance dominates at ~280 nm[44,45]. The absorbance spectra exhibited glucose-dependent color switching in 12- or 24-point titration series (Fig. 3a, b; Supplementary Fig. 5). Above 300 nm, fluorophore absorbance dominates, and SVD analysis revealed four distinct regions (Fig. 3; Supplementary Fig. 5), three of which (360 ± 10 nm, 380 ± 10 nm, 420 ± 10 nm) were present in all tested conjugates, and one of which (320 ± 10 nm) was present in ecGBP.16C•Badan (Fig. 3c) and ttGBP.17C•Badan (Supplementary Fig. 5). The hypso- or bathochromic directions of absorbance switching correlated with corresponding emission color switches (*cf*. Fig. 2b with Fig. 3a, and Fig. 2c with Fig. 3b). In all conjugates, the 380 and 420 nm regions changed sign in $C_2$, indicating that their exchange dominates two-state color switching in the ground state. The $C_2$ contribution also was accurately modeled with the two-state glucose-binding model (Fig. 3a, b, insets). We therefore inferred

**Table 2 Structure determination by X-ray crystallography.**

|  | ecGBP.W183C•Acrylodan | ttGBP.F16C•Badan |
|---|---|---|
| Data collection |  |  |
| Wavelength (Å) | 1.000 | 1.000 |
| Space group | C121 | P2$_1$2$_1$2$_1$ |
| Unit cell parameters |  |  |
| a, b, c (Å) | 119.58, 36.53, 79.90 | 45.49, 53.06, 134.32 |
| α, β, γ (°) | 90.00, 124.13, 90.00 | 90.00, 90.00, 90.00 |
| Resolution (Å) | 50.00 - 1.53 | 41.63 - 1.59 |
|  | (1.56 - 1.53) | (1.65 - 1.59) |
| $R_{merge}$ | 0.06 (0.29) | 0.09 (0.74) |
| $R_{pim}$ | 0.025 (0.131) | 0.034 (0.29) |
| Half-set correlation CC(1/2) | 0.993 (0.956) | 0.999 (0.835) |
| $I / \sigma I$ | 25.54 (3.99) | 16.44 (3.26) |
| Completeness (%) | 98.8 (94.6) | 99.83 (98.72) |
| Redundancy | 6.5 (5.1) | 7.2 (7.2) |
| Refinement |  |  |
| No. reflections | 42706 | 44517 |
| $R_{work}$ / $R_{free}$ | 0.1558/0.1827 | 0.1608/0.1775 |
| No. atoms |  |  |
| Protein | 2358 | 2471 |
| Acrylodan/Badan | 17 | 16 |
| Glucose | 72 | 12 |
| Ion | 1 | 4 |
| Water | 318 | 407 |
| B-factors |  |  |
| Protein | 15.69 | 17.24 |
| Acrylodan/Badan | 40.18 | 35.92 |
| Glucose | 21.41 | 10.63 |
| Ion | 9.85 | 32.23 |
| Water | 33.01 | 29.51 |
| R.m.s. deviations |  |  |
| Bond lengths (Å) | 0.072 | 0.008 |
| Bond angles (°) | 1.92 | 1.05 |

$C_2$ to encode excitation of the dominant blue↔green exchanging emissions. In some spectra (Supplementary Fig. 5), small $C_2$ regions with opposing sign also were present at 320 nm and 360 nm, which we inferred to mark minor violet↔cyan emission exchanges. In conjugates where this exchange is absent, only the 360 nm absorbance was observed.

**The structures of the green- and blue-emitting ground states differ in fluorophore carbonyl group twisting.** We determined high-resolution X-ray structures representing the ground states of the two major spectral forms: the glucose complexes of the Blue ttGBP.F17C•Badan (Supplementary Data 1) and Green ecGBP.W183C•Acrylodan (Supplementary Data 2) conjugates (Table 2). The conjugated fluorophores are clearly visible in the electron densities, albeit at lower occupancies than the protein residues, enabling chromophore models to be built (see Supplementary Figs. 7 and 8 for simulated annealing omit maps). In ttGBP.17C•Badan the fluorophore is attached to the N-terminal domain, whereas in ecGBP.W183C•Acrylodan it is attached to the C-terminal domain. The overall structures of ecGBP.W183-C•Acrylodan and ttGBP.17C•Badan are the same as the unconjugated, wild-type ecGBP closed conformation of its glucose complex (Fig. 4)[41].

The aromatic groups that are replaced by the two conjugates contact the opposite sides of the glucose pyranose ring (Fig. 1a). The conjugated Acrylodan and Badan naphthalene rings do not replace the original aromatic contacts. Instead, the rings point out of the glucose-binding pocket into the solvent and are each wedged inside a narrow channel ("aromatic channel") formed within the interface between the N- and C-terminal domains (Fig. 4a–c). Each conjugate protrudes through a different channel, located in different parts of the interface, ~120° apart (Fig. 4a).

The rotatable carbonyls are buried entirely within each protein, and grasped by two (ttGBP.17C•Badan, Fig. 4d–f) or three (ecGBP.183C•Acrylodan, Fig. 4g–i) residues within small pockets ("carbonyl holes") located adjacent to the mutant cysteine to which their conjugate is attached. In the ecGBP.W183C•Acrylodan glucose complex, the ground-state carbonyl is coplanar with the naphthalene ring and emits a green color when excited (Fig. 4g, h), whereas in the ttGBP.F17C•Badan glucose complex (Fig. 4d, e) it has twisted ~30° out of plane, and emits a blue color upon excitation. These structures therefore establish that the dominant two colors correlate with a carbonyl twist in the ground state.

The residues comprising the carbonyl holes are divided across the interface between the two domains. Some interact both with the conjugate and the bound glucose (Fig. 4f, i). In ttGBP.F17C•Badan, the carbonyl hole is formed by D238 and N258. D238 lies within the C-terminal domain and forms hydrogen bonds with the glucose 2- and 3-hydroxyls. N258 is located on one of the hinge strands, and forms a hydrogen bond with the glucose 1-hydroxyl. The $C_\beta$ and $C_\gamma$ atoms of both residues form van der Waals contacts with the twisted carbonyl. In ecGBP.W183C•Acrylodan, the carbonyl hole is formed by K92, E149, and H152 contacts. E149 and H152 are located in the C-terminal domain, whereas K92 lies within the N-terminal domain. The H152 imidazole $N_\epsilon$ forms a hydrogen bond with the glucose 6-hydroxyl; the $C_\delta$ atom forms a van der Waals contact with the planar carbonyl. K92 and E149 do not contact glucose, but form a salt bridge across the interdomain interface in the closed conformation; the K92 $N_\zeta$ forms a hydrogen bond with the carbonyl. The carbonyl holes therefore change in response to glucose-mediated transitions between the open and closed protein conformations.

By contrast, the dimethyl amino (DMA) groups of the fluorophores in these two structures point into solution and are isolated from the surrounding protein (Fig. 4d, g). Consequently, these rotatable DMAs are unconstrained and adopt their relaxed, planar conformation[46].

**Protein-carbonyl interactions determine direction and extent of color switching.** The attachment position can affect the direction of the color switch: ecGBP.F16C and ttGBP.F17C conjugates switch in the opposite direction (Green→Blue) from their ecGBP.W183C and ttGBP.W182C (Blue→Green) counterparts upon glucose addition (Fig. 2). The tryptophan residues of the *E. coli*, *T. thermoanaerobacter*, and *G. klaustophilus* GBP homologs align[34], but gkGBP.W168C•Acrylodan switches in the direction opposite to the equivalent ecGBP.W183C and ttGBP.W182C conjugates (Supplementary Fig. 6). The direction of color change therefore is not determined by the nature of the replaced amino acid (F or W). In ttGBP.W182C, the Acrylodan, but not the Badan conjugate switches (Fig. 2c). Differences in linker length therefore can affect switching (conjugated Acrylodan is one methylene unit longer than Badan; Fig. 1c, d). Taken together, these results show that the direction and magnitude of switching is critically dependent on the precise interactions between the placed fluorophore and the protein near the conjugate attachment point, which is the region where the carbonyl hole is formed.

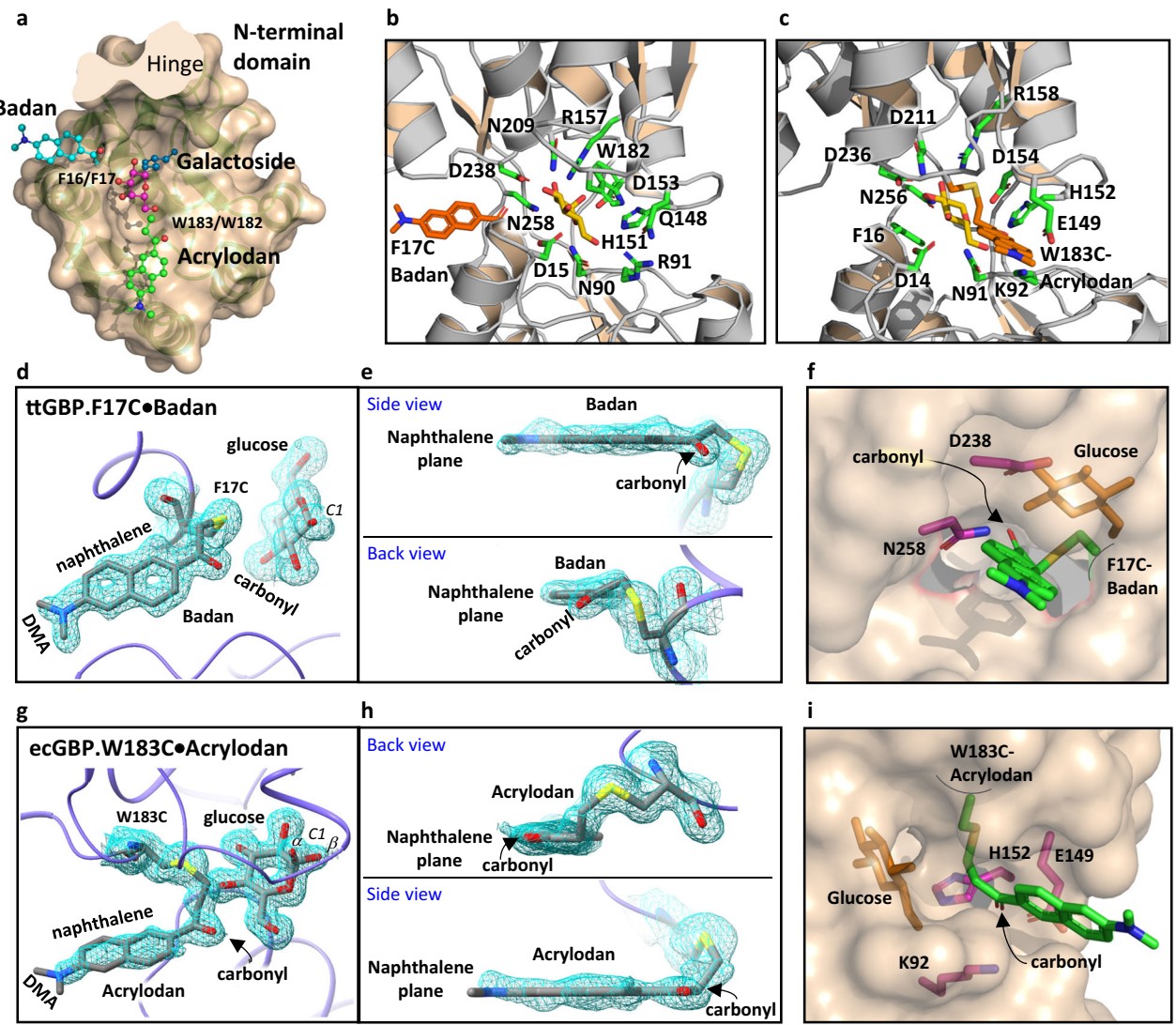

**Fig. 4 Structures of planar and twisted fluorescent conjugates bound to glucose. a** Superimposition of the ecGBP and ttGBP N-terminal domains (the C-terminal domain was cut away at the hinge) illustrates that the naphthalene rings of the ecGBP.W183C•Acrylodan (green; navy, nitrogen; red, carbonyl) and ttGBP.F17C•Badan (blue) conjugates are positioned in two different channels such that their DMA groups protrude into the solvent. The conjugates are anchored at cysteine mutations (F16C or F17C; W183C or W182C) located adjacent to the bound glucose (magenta). A galactoside *R* group (navy, but not present in these structures) protrudes into a third channel[42]. **b** Amino acid side-chains (green) that interact with the bound glucose (yellow) in the ttGBP.F17C•Badan (orange) complex. **c** The glucose-binding site in the ecGBP.W183C•Acrylodan complex. **d, e** The $2F_o$-$F_c$ electron density map (0.7 Å contours) of Badan and glucose in ttGBP. F17C•Badan (gray, carbon; red, oxygen; blue, nitrogen; yellow, sulfur). The fluorophore carbonyl is twisted out of the naphthalene plane in this complex which exhibits a Blue emission spectrum. **f** The carbonyl hole is formed by N258 and D238 contacts. **g, h** The $2F_o$-$F_c$ electron density map (0.7 Å contours) of Acrylodan and glucose (there are two 1-hydroxyl epimers in the density) in ecGBP.W183C•Acrylodan. The fluorophore carbonyl is coplanar with the naphthalene plane in this complex which exhibits a Green emission spectrum. **i** The carbonyl hole is formed by K92, E149, and H152 contacts. PDB accession codes: 8fxt, ecGBP.W183C•Acrylodan (green); 8fxu, ttGBP.F17C•Badan (blue).

**Shifts in conformational equilibria between planar and twisted fluorophores**. We mutated the residues forming carbonyl hole contacts in the Acrylodan and Badan conjugates of ecGBP.F16C, ttGBP.F17C, ecGBP.W183C and ttGBP.W182C (Fig. 4f, i), and measured their emission spectra in glucose titrations (Table 1, Supplementary Figs. 1–4). The majority of the 47 mutations retained glucose binding (with a range of affinities) and color-switching directions. However, their apparent colors exhibited considerable, systematic variation. The apparent color change, $\Delta\lambda_B$, and logarithm of the relative intensity change, $\rho^*$, are linearly related in the mutants to a first approximation (Fig. 5a). Such a linear relationship is consistent with a dominant two-state equilibrium in which the direction and extent of color switching lie on a continuum corresponding to distinct distributions of planar, Green, and twisted, Blue fluorophores with concomitant apparent colors in the open (apo) or closed (glucose-bound) protein conformations. In this mechanism, colors may be modulated additionally to a lesser extent by other interactions between the chromophore and surrounding residues. Such additional effects may account for the difference between the Yellow and Green forms. Glucose binding shifts the two-state protein population from open to closed conformations with a concomitant change in the apparent color (Fig. 5b). The larger the difference between the fluorophore conformational twisting equilibria in the open and closed protein conformations, the larger the color switch. Conversely, if the two twisting equilibria are approximately equal, the

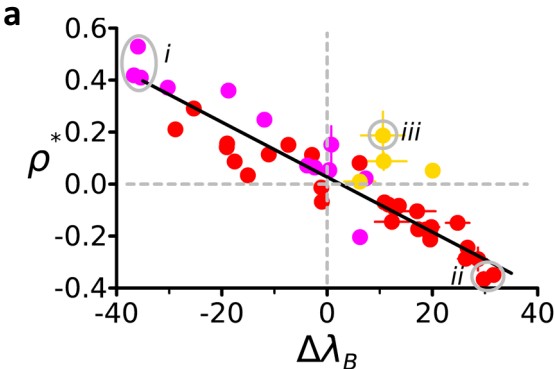

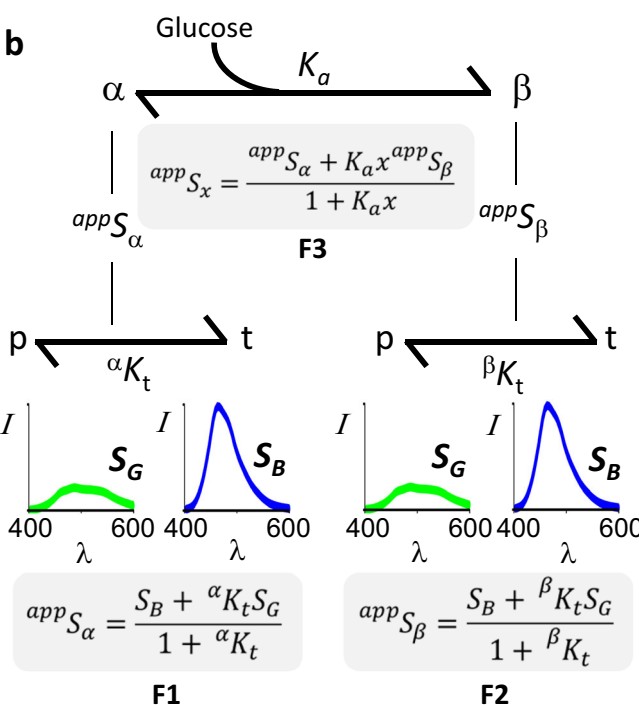

**Fig. 5 Equilibrium distributions of spectral states. a** Mutations in residues that interact with the carbonyl of Acrylodan and Badan conjugates give rise to a linear relationship between the extent of Blue↔Green color switching as monitored by the change in the barychrome, $\Delta\lambda_B$, and logarithm of the relative intensity change $\rho^*$ (red circles: Acrylodan conjugates; magenta circles, Badan conjugates). Gray circles mark the most effective glucose responses: $i$, Green→Blue ttGBP.17C•Badan and its N258A and N258L mutants; $ii$, Blue→Green ttGBP.183C.R91K.Q148E•Acrylodan and ttGBP.183C.R91S•Acrylodan. The Green→Yellow switch of ecGBP.183C Badan and its mutants (yellow circles) exhibit a constant, small increase in relative intensity, regardless of extent of color switching with the exception of the ecGBP.183C.H152N•Badan mutant ($iii$). Horizontal and vertical lines show the 1σ distribution of the $\Delta\lambda_B$ and $\rho^*$ parameters, respectively, determined in 10,000 bootstrap trials. **b** Statistical thermodynamic model for linking ligand-mediated protein conformational changes to chromophore twisting. Prodan derivatives can assume either a planar, $p$, or a twisted, $t$, carbonyl conformation, associated with Green, $S_G$, and Blue, $S_B$, emission spectra, respectively. These fluorophore forms are in equilibrium $p \rightleftharpoons t$ with equilibrium constants $^{\alpha}K_t$ and $^{\beta}K_t$ in the apo and glucose-bound protein, respectively, forming an equilibrium distribution with concomitant apparent emission spectra $^{app}S_{\alpha}$ and $^{app}S_{\beta}$ from a mixture of Green and Blue spectra (equations F1 and F2). If protein-carbonyl interactions favor twisting, $^{\alpha\,or\,\beta}K_t \to \infty$ and the apparent emissions are Blue, $^{app}S_{\alpha\,or\,\beta} \to S_B$; conversely, if the planar form is stabilized, $^{\alpha\,or\,\beta}K_t \to 0$ and $^{app}S_{\alpha\,or\,\beta} \to S_G$. If the torqueing interactions do not establish these equilibrium extremes, then an intermediate color is observed in the open or closed protein conformation(s). Glucose-dependent colors, $^{app}S_x$, arise from changes in the relative populations of $^{app}S_{\alpha}$ and $^{app}S_{\beta}$ due to ligand binding (equation F3; $K_a$ is the glucose-binding affinity, $K_d = 1/K_a$, and $x$ the glucose concentration; note F3 requires substituting F1 and F2 for its $^{app}S_{\alpha}$ and $^{app}S_{\beta}$ terms, respectively). If $^{\alpha}K_t<^{\beta}K_t$, then the color switches Green→Blue; if $^{\alpha}K_t>^{\beta}K_t$, Blue→Green. The greater $|\log(^{\alpha}K_t/^{\beta}K_t)|$, the more effective the color switch, as is the case at either end of the distribution in panel **a**.

conjugate is "stuck", and no color switch is observed (e.g. ttGBP.W182C•Badan).

Carbonyl hole mutations can optimize color switching by altering fluorophore conformational equilibria. For instance, the D238N mutation in the carbonyl hole of ttGBP.F17C•Acrylodan improves color switching from ($\Delta\lambda_B = -15$ nm, $\rho^* = 0.03$) to ($\Delta\lambda_B = -29$ nm, $\rho^* = 0.21$) (Table 1). At either end of the observed distribution, the twisting equilibrium adopts extreme values such that one chromophore conformation dominates, with its concomitant color (Fig. 5 groups i and ii). Carbonyl holes that stabilize such extremes encode optimal color-switching biosensors.

## Conclusions

**Color switching arises from protein-mediated shifts in the equilibrium distribution of two fluorophore conformations.** We determined that the Acrylodan and Badan derivatives of three GBP homologs switch between two predominant emission spectra with apparent intense Blue, and less intense Green (occasionally Yellow) colors. Bidirectional switching of the emission spectra established that either color can exist in the apo and glucose-bound proteins, corresponding to open and closed protein conformations. Absorbance spectra also undergo color

switching, in the same direction as associated emission spectra, from which we conclude that ground-state effects play a key role in determining emission colors.

Two X-ray crystal structures of glucose complexes revealed that the carbonyl of Prodan derivatives forms either a planar or a twisted ground-state conformation relative to the naphthalene ring, associated with Green and Blue excited-state emission spectra, respectively. The fluorophore carbonyls are buried in small protein holes, adjacent to the fluorophore linker attachment site. The carbonyls are contacted by two or three residues located on different domains whose relative disposition changes in the open and closed protein conformations. In the closed protein conformation, the naphthalene ring protrudes out of the protein through a channel comprising residues contributed by the N- and C-terminal protein domains. These interactions therefore also are sensitive to the two protein conformational states. We therefore conclude that if these two sets of interactions are different relative to each in the open (apo) and closed (glucose-bound) conformations, the carbonyl "torques" in or out of the naphthalene plane, switching color. This arrangement also links apparent colors to the glucose-binding equilibrium, because ligand binding shifts the distribution of protein conformations with a concomitant effect on the global fractions of their embedded Green and Blue fluorophore conformations in this system of conformationally coupled equilibria (Fig. 5b)[29].

Carbonyl hole mutations revealed that the apo and glucose complexes each can adopt a continuum of apparent colors. Open and closed protein conformations therefore embed distinct equilibrium distributions of mixed planar Green and twisted Blue fluorophores with concomitant apparent colors.

Consequently, the magnitude of the glucose-mediated color switch is determined by the degree of bias in the fluorophore conformational equilibria embedded within each protein conformational state.

The relaxed Prodan ground-state carbonyl is planar[46]. Torqueing interactions can be expected to be less constraining in the open than in the closed protein conformations. Accordingly, the conjugated fluorophore would be expected to be planar and Green in open states. However, we observed both Green→Blue and Blue→Green color switching in the open→closed transition; under the right combination of circumstances, either protein conformation therefore provides interactions near the fluorophore attachment site that can torque the fluorophore.

**Functional group twisting**. Prodan was originally designed as a push-pull chromophore in which the photo-excited electron transfers from the dimethyl amino (DMA) lone pair to the carbonyl thereby establishing a large, charge-transfer transition dipole in the excited state[47]. As intended, Prodan emissions are strongly dependent on solvent polarity. A commonly accepted view is that Prodan derivatives report protein conformations via solvatochromic effects that give rise to color switching due to changes in the dielectric of their local environment[44,45,48–51]. An alternative model invokes DMA twisting in the excited state[52,53]. Our experimental observations revealed that ground-state carbonyl twisting controls color switching. The electronic structures of the planar and twisted conformations in their ground and excited states, and the mechanism by which these give rise to color switching are currently under investigation.

We found that in color-switching Acrylodan and Badan conjugates, the blue↔green exchange dominates, because their linkage chemistry places the twistable carbonyl of Prodan within pockets where it contacts protein residues directly. Examination of published spectra of other biosensors based on Acrylodan or Badan conjugates revealed similar exchanges[9–22]. By contrast, the linker chemistry of Anap conjugates, places the twistable amine of Prodan in close proximity to the protein, (Fig. 1e), whereas their twistable carbonyl is unlikely to form contacts (i.e. the obverse of Acrylodan or Badan coupling). For Anap conjugates, a violet↔cyan dual emission dominates color switching whereas the green↔blue exchange is absent[10,11]. These observations suggest that both carbonyl and amine Prodan groups can twist to elicit color switching, and that linker stereochemistry determines the dominant dual emission by selecting the twistable group that is torqued by protein conformational changes. Indeed, some of the Badan conjugates presented here show a minor contribution of the violet↔cyan exchange (Fig. 2d, Table 1), presumably because their twistable amine (DMA) is placed close to the protein surface, forming contacts that influence its twist.

**Solute effects**. We found that solvatochromic effects remain present in the Prodan conjugates, but they do not dominate glucose-mediated color switching. In the high-concentration phase of the biphasic glucose titrations, all conjugates undergo small hypsochromic shifts and intensity increases, regardless of apparent color. The refractive index of glucose solutions increases with solute concentration[54], and their dielectric constant decreases[55,56], due to decreased solvent polarizability through solvent mobility loss by trapping mobile bulk water molecules within the glucose solvation sphere. The high-glucose responses therefore are consistent with this water mobility loss causing hypsochromic shifts via solvatochromic effects, and increased emission intensities via decreased collisional quenching with water.

**Structure-based design of color-switching biosensors by torqueing functional groups**. The torqueing model presented here provides a conceptually straightforward structure-based framework for designing biosensors: identify ligand-responsive backbone positions[57] that enable emplacement of fluorophore rings into aromatic channels and twistable functional groups into holes that undergo ligand-mediated conformational change, and which either exist fortuitously or are created by protein design. Optimal color switching is achieved when apo and ligand-bound complexes each stabilize a different, unique fluorophore conformation (planar or twisted). This approach will enable creation of next-generation in vitro and in vivo reagents for cellular imaging, structure-function studies, and biosensing with clinical and environmental applications.

## Materials and methods

**Preparation of fluorescent conjugates**. Procedures for protein expression, purification, fluorophore labeling and mutagenesis have been described previously[34]. Acrylodan and Badan were purchased from ThermoFisher Scientific.

**Measurement of fluorescence emission and absorbance spectra**. Fluorescence emission spectra were collected on a Nanodrop 3300 spectrofluorimeter (ThermoFisher Scientific) using 2 μL droplets with 10 μM protein. Absorption spectra were collected on a Nanodrop 2000 spectrophotometer (ThermoFisher Scientific), using 50–56 μM conjugate protein. Glucose titrations (12–24 data points) were carried out at room temperature in 20 mM MOPS (pH 7.4), 20 mM KCl, 1 mM CaCl$_2$.

**Structure determination by X-ray crystallography**. ecGBP.183C•Acrylodan or ttGBP.17C•Badan were mixed with 2mM D-glucose and 1 mM CaCl$_2$. Sitting-drop vapor-diffusion crystallization trials were carried out at 17 °C using sparse-matrix screening conditions (Hampton Research). Both conjugates crystallized in 20% PEG 3350 and 0.2 M potassium thiocyanate as clusters of needles. Single crystals were isolated mechanically (micro-tools, Hampton Research), transferred stepwise into mother liquor containing 30% ethylene glycol, and flash-frozen in liquid nitrogen. Diffraction data were collected remotely at the Advanced Photon Source, SER-CAT beamline 22-ID. The ecGBP.183C•Acrylodan and ttGBP.17C•Badan crystals diffracted to 1.39 Å and 1.59 Å resolution, respectively. The data was phased by molecular replacement using PHASER[58] with a poly-alanine backbone of ecGBP (2gbp)[41] as the search model. Multiple rounds of positional, individual B-factor and occupancy refinement and subsequent model building were carried out in PHENIX[59] and COOT[60]. Solvent molecules were added both automatically (phenix.refine) and manually. The structures were validated using PHENIX tools. Omit maps were generated by leaving out the coordinates for the conjugated Badan or Acrylodan (but not the cysteine to which they are attached) from the final model, followed by three rounds of refinement by simulated annealing using PHENIX.

**Numerical analysis**. The methods described here are implemented as Python scripts that use components of the Biomolecular Computing (BMC) package developed by HWH. These scripts and the BMC package can be downloaded from the Dryad site (see Data Availability statement).

**Modeling ligand binding with two-state isothermal binding polynomials**. The color shifts were described using the intensity-

weighted dominant emission "barychrome" wavelength[34], $\lambda_B$,

$$\lambda_B = \frac{\sum I_\lambda \lambda}{\sum I_\lambda} \quad (1)$$

where $\lambda$ is the wavelength, and $I_\lambda$ its corresponding absolute emission intensity; the sums are calculated over wavelength $[\lambda_{min}, \lambda_{max}]$ where $I_\lambda > 0$. $\Delta\lambda_B$ refers to the difference of the $\lambda_B$ values of the saturated glucose complex relative to the apo-protein.

A given glucose response signal, $S$ (e.g. $\lambda_B$), can be fit with a ligand-binding reaction as

$$S = \alpha(1 - \bar{y}) + \beta\bar{y} \quad (2)$$

where $\alpha$ and $\beta$ are signal baselines associated with the apo- and glucose-bound protein, respectively, and $\bar{y}$ is the fractional occupancy for a single glucose-binding site[29]

$$\bar{y} = \frac{K_a L}{1 + K_a L} \quad (3)$$

with glucose concentration $L$, and affinity $K_a$ (dissociation constant $K_d = 1/K_a$). A baseline $B$, ($\alpha$ or $\beta$ in Eq. 2), can be a glucose-independent constant or exhibit a dependence on glucose that is linear

$$B = B_0 + B_1 L \quad (4)$$

Binding models were fit to experimental data using non-linear least-squares methods as described[34].

**Relative intensity changes**. The fitting methods produce a set of spectra whose intensities are self-consistently scaled from which intensity changes relative to the intensities of the emission spectrum of the glucose-free conjugate can be calculated. The wavelength-dependent relative intensity change, $\rho_\lambda$, is given by

$$\rho_\lambda = \log \frac{I_\lambda}{I_{\lambda,0}} \quad (5)$$

where $I_\lambda$ is the emission of the spectrum at wavelength $\lambda$, and $I_{\lambda,0}$ the corresponding intensity in the apo-protein emission spectrum. The logarithm of the ratio is used to place increases and decreases on the same scale, different only by sign. The relative integrated intensity of an entire emission spectrum is given by

$$\rho = \log \frac{\sum_\lambda I_\lambda}{\sum_\lambda I_{\lambda,0}} \quad (6)$$

where the summation is for all wavelengths at which $I_\lambda > 0$. $\rho^*$ refers to the change in values of the glucose complex relative to the apo-protein.

**Estimates of parameter and model fit uncertainties**. We used bootstrapping to compute distributions of parameter values and curve fits using 10,000 trials[29,61]. Briefly, in each bootstrap trial, a randomly chosen subset of 37% of the experimental data-points were duplicated, and the model was refit. Parameter values were then estimated as the mean of their computed values, and uncertainties as their standard deviations. The distribution of the calculated models is shown as the standard deviations of the calculated observations at each experimental data point.

**Identifying dominant spectral changes by singular value decomposition**. The singular value decomposition method[43,61] was used to determine spectral features for describing the ligand-mediated spectral shape changes in a titration series. Using the self-consistently scaled spectra, we constructed a $m \times n$ matrix, $A$, where $m$ is the number of wavelengths, and $n$ the number of titration points. The entries in $A$ are intensities, with each column in $A$ corresponding to the emission spectrum at a particular ligand concentration, and each row to the intensities at a particular wavelength across all ligand concentrations in the titration series. It has been shown that $A$ can be decomposed into three matrices

$$A = USV^T \quad (7)$$

where $U$ is another $m \times n$ matrix, $S$ is a diagonal $n \times n$ matrix, and $V$ an $n \times n$ matrix. Each column $i$ in $U$ represents a spectral feature or "component" vector $C_i$ with an intensity contribution (which can be positive or negative) ordered by wavelength. The columns are ordered such that $C_1$ corresponds to the component that by itself accounts for most of the data in $A$, and component $C_n$ the least. The $V$ matrix records the contribution of the components at each titration point; it preserves the ordering of the ligand titration concentrations in the titration series. The $S$ diagonal elements, $s_{i,i}$, record the global weight of each component at all titration points. The ordering of the information in $U$ also ensures that these elements obey $s_{i+1,i+1} \geq s_{i,i}$.

The fractional weight, $f_i$, of the $i^{th}$ component is given by

$$f_i = \frac{s_{i,i}}{\sum_{j=1}^{j=n} s_{j,j}} \quad (8)$$

The contribution of the $i^{th}$ component at each ligand concentration in the titration series is given by the $i^{th}$ row or column of $V$. A vector corresponding to a row (or column) in $V$ therefore represents the dependency of a component on ligand concentration and its elements can be treated as a signal $S$ in Eq. 2. The signs of the elements in $U$ and $V$ are coupled and degenerate (e.g. -,+ is equivalent to +,-). We therefore chose signs such that $v_1 < v_n$ ($v_1$ and $v_n$ are the first and last elements in $V$, respectively) to represent monotonic increases in component contributions. The ligand dependencies of multiple components share the same affinity, $K_a$, but each have a distinct pair of baselines $\alpha_i$, $\beta_I$, and can be combined into a global, non-linear fit.

Only the top components in $S$ encode glucose-dependent information, the rest encode experimental noise. Two criteria were used to determine the boundary between these two sets. First, we used Pearson's correlation[43] to determine whether the elements in $V_i$ correlate monotonically with glucose concentration, eliminating all vectors $V_{j \geq i}$ if $V_i$ had a correlation $p$ value $p > 0.01$, or if $V_i$ could not be fit with Eq. 2.

**Electronic data and software**. The spectral and glucose-binding data, Python scripts used for analysis, the BMC software package, and calculation results reported here can be downloaded from https://doi.org/10.5061/dryad.msbcc2g2x.

## Data availability

Accession codes. Coordinates and structure factors have been deposited in the Protein Databank with accession codes 8FXU, ttGBP.F17C•Badan (Supplementary Data 1) and 8FXT, ecGBP.W183C•Acrylodan (Supplementary Data 2).

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

## Acknowledgements

We thank Andrew Bergeron and Jayalakshmi Miriyala for assistance with a subset of protein preparations and spectroscopic data collection, and Elizabeth McSweeney for X-ray diffraction data collection of ecGBP.183C•Acrylodan. X-ray data were collected at Southeast Regional Collaborative Access Team (SER-CAT) 22-ID (or 22-BM) beamline at the Advanced Photon Source, Argonne National Laboratory. SER-CAT is supported by its member institutions, and equipment grants (S10_RR25528, S10_RR028976 and S10_OD027000) from the National Institutes of Health. Use of the Advanced Photon Source was supported by the U. S. Department of Energy, Office of Science, Office of Basic Energy Sciences, under Contract No. W-31-109-Eng-38. Data were also collected at Beamline 8.3.1 of the Advanced Light Source, a U.S. DOE Office of Science User Facility under Contract No. DE-AC02-05CH11231, is supported in part by the ALS-ENABLE program funded by the National Institutes of Health, National Institute of General Medical Sciences, grant P30 GM124169-01. Structural biology applications used in this project were compiled and configured by SBGrid. This work was supported in part by Corporate Research Agreements between Duke University and Becton-Dickinson and Company (Research Triangle Park, North Carolina, USA; Duke University Contract AGR20112333), and SenGenix Inc. (Durham, North Carolina, USA), and by the Department of Biochemistry.

## Author contributions

Experimental design: H.W.H. and M.J.A. Protein chemistry: M.J.A. Spectroscopy: M.J.A. Data analysis: M.J.A. and H.W.H. Software and modeling: H.W.H. X-ray crystallography: S.K., Y.W., and L.S.B. M.J.A. and H.W.H. wrote the manuscript.

## Competing interests

Homme W. Hellinga and Malin J. Allert are co-inventors on two patents concerning protein sequences, crystal structures, and fluorescent emission studies of proteins described in this work: "Glucose/Galactose Biosensors and Methods of Using Same", US 11,352,657 B2 (issued June 7, 2022; region, USA; assignee, Duke University Medical Center); "Glucose Biosensors and Uses Thereof", US 11156615 B2 (issued Oct 26, 2021; region, USA; assignee, Duke University Medical Center). All other authors declare no competing interests.
