## [Peer Review File · Communications Chemistry]

Reviewers' comments:

Reviewer #1 (Remarks to the Author):

In this paper, the group of H. W. Hellinga presents a fluorescent glucose biosensor based on the fluorogenic/solvatochromic properties of the fluorophore Prodan. Beyond the development of this sensor, the work includes a comprehensive study-the most complete I have seen-of the physicochemical basis of the emission changes through analysis of two x-ray structures of conjugates with blue and green emission and mutagenesis studies, which provide compelling evidence for the fluorogenic mechanism of Prodan through carbonyl twisting, which provides a new conceptual framework for the development of this type of sensors. The study is highly relevant for those interested in the development of solvatochromic fluorophore-based biosensors and for researchers in the field of protein engineering.

I recommend the publication of this article with minor changes.

- Clearly, the x-ray structures and mutagenesis studies support the influence of carbonyl twist on the spectral changes (coplanar-green, twisted-blue), but it would be interesting to have electronic calculations relating these structures to electronic levels, at least a preliminary calculation of the two fluorophore conformations in the absence of the protein, to have a more robust view of the rationale for these experimental observations.

- A description of the numerical analysis is included in the Materials and Methods section and it is indicated that it was performed using Python scripts based on the Biomolecular Computing package developed by HWH. Would it be possible to give more details of these scripts and the software used?

- If the researchers do not deposit the x-ray structures in the PDB database, it is essential that these structures are included as part of the supplementary information, as they are fundamental to understanding this article.

Reviewer #2 (Remarks to the Author):

Review for: Chromophore carbonyl twisting in fluorescent biosensors encodes direct readout of protein conformations with multicolor switching

Overall: This is a fairly complex paper, but is written simply and clearly. The results is impactful and novel. The quality of the science is generally very high, particularly the spectroscopy. I could find no fault in the interpretation. We shall see how widely taken up the approach will be - dependence on Cys tagging remains a pain. Perhaps in the discussion some mention of the limitations could be made. Overall, an interesting and good read - not too much to complain or argue about. That being said, I am not expert-expert in this type of system - I would defer to a more qualified spectroscopist if one of the other reviewers has an issue. I can comment on the

structural biology and engineering, both of which are good.

I have a few minor suggestions that could improve the paper (perhaps):

Figure 1A - protein structure panel too small and zoomed out - needs a close up view of the active site and mutated (modified positions).

Figure 4: Should show unbiased omit density. I am concerned that 0.7 sigma 2mfo-dfc is perhaps affected by model bias.

SI Table S1: Should include CC1/2

Reviewer #3 (Remarks to the Author):

The authors describe a strategy wherein they use thiol-reactive versions of the naphthalene fluorophore Prodan interfaced within the glucose binding site of periplasmic glucose binding protein. Specific residues that bind glucose were mutated to cysteines to enable residue-specific modification. Glucose binding results in color emission switching across four color channels that tracks with the twisting rotation of the fluorophore carbonyl bond with respect to the plane of the naphthalene ring. At different glucose concentrations, the color emission changes from the distribution of two spectral states. This approach represents a new way of designing recombinant protein biosensors that are coupled to ligand-induced conformational changes in the host protein binding site that translate into quantitative spectroscopic changes. I find the work to be of high quality and exhaustively performed and clearly presented. I see no areas in which to improve the study or the manuscript as presented. I recommend the manuscript for publication in its current form.

Response to Referees' comments, July 17, 2023; MS: COMMSCHEM-23-0282.

We thank the referees for their time and comments. Our response is given below.

Referee #1

Query: "Clearly, the X-ray structures and mutagenesis studies support the influence of carbonyl twist on the spectral changes ..., but it would be interesting to have electronic calculations relating these structures to electronic levels, at least a preliminary calculation of the two fluorophore conformations in the absence of protein, to have a more robust view of the rationale for these experimental observations."

Response: We agree with the referee that a mechanistic understanding of the observations in terms of underlying electronic structures would be interesting. In fact, we have undertaken a major effort to achieve that goal. It turns out that the quantum effects are complex, and our experiments indicate that these appear to involve an exchange between singlet and triplet states as a consequence of carbonyl twisting. These effects are challenging both to probe experimentally, and to model with electronic structure calculations. This study is too complex to include in this manuscript, and will be the subject of another publication. "Simple" ground-state calculations requested by the referee would be highly misleading. To acknowledge that electronic structures are of obvious interest, we added a sentence in the Conclusions section of the manuscript: "The electronic structures of the planar and twisted conformations in their ground and excited states, and the mechanism by which these give rise to color switching are currently under investigation" P 13, bottom paragraph 3.

Query: "A description of the numerical analysis is included in the Materials and Methods section and it is indicated that it was performed using Python scripts based on the Biomolecular Computing package developed by HWH. Would it be possible to give more details of these scripts and the software used?"

Response: The protocols implemented by the scripts are detailed in the Materials and Methods. The scripts and the entire software package that they are part of are downloadable from the Dryad repository (the link was provided in the original submission, including the private link accessible to reviewers in "ASSOCIATED CONTENT"). In Materials and methods, we have added a clarifying statement

"These scripts and the BMC package can be downloaded from the Dryad site (see Data Availability statement)."

This information is given in the Data Availability statement.

Query: "If the researchers do not deposit the x-ray structures in the PDB database, it is essential that these structures are included as part of the supplementary information, as they are fundamental to understanding this article."

Response: Actually, both structures were deposited prior to submission (pdb 8fxt, 8fxu), as reported in the manuscript under "ASSOCIATED CONTENT". To make this more obvious, we have also added these two accession codes to the legend of figure 4 in the revised MS (last sentence). We also have uploaded the two PDB structure reports to journal's website.

Referee #2

Query: “We shall see how widely taken up the approach will be - dependence on Cys tagging remains a pain. Perhaps in the discussion some mention of the limitations could be made.”

Response: We agree that such a discussion would be interesting. However, this article is not an appropriate forum to review the pros and cons of thiol-mediated conjugation strategies, non-natural amino acid incorporation, or fluorescent protein fusions, to mention a few. Nor would such a review provide insight into the mechanism under investigation.

Query: “Figure 1A - protein structure panel too small and zoomed out - needs a close up view of the active site and mutated (modified positions).”

Response: We have enlarged panel 1a. The mutated positions are shown in Fig. 4b, c, f, and i.

Query: “Figure 4: Should show unbiased omit density. I am concerned that 0.7 sigma 2mfo-dfc is perhaps affected by model bias.”

Response: We have added the following sentence to main text (p. 8). “The conjugated fluorophores are clearly visible in the electron densities, albeit at lower occupancies than the protein residues, enabling chromophore models to be built (see Figs. S7 and S8 for simulated annealing omit maps).” We have shown unbiased omit densities in new, additional supplementary figures Fig. S7 and S8. We added the following sentence to Materials and Methods (bottom p. 16): “Omit maps were generated by leaving out the coordinates for the conjugated Badan or Acrylodan (but not the cysteine to which they are attached) from the final model, followed by three rounds of refinement by simulated annealing using PHENIX.”

Query: “SI Table S1: Should include CC1/2”

Response: The statistic has been added to table S1.

Referee #3

Positive reinforcement; no response needed.